# Precise automatic classification of 46 different pollen types with convolutional neural networks

Víctor Sevillano[1], Katherine Holt[2], José L. Aznarte[1]*

**1** Artificial Intelligence Department, Universidad Nacional de Educación a Distancia—UNED, Madrid, Spain, **2** Institute of Natural Resources, Massey University, Palmerston North, New Zealand

* jlaznarte@dia.uned.es

**Data Availability Statement:** The data underlying this study is available via Figshare: https://doi.org/10.6084/m9.figshare.12370307.

**Funding:** The authors received no specific funding for this work.

## Abstract

In palynology, the visual classification of pollen grains from different species is a hard task which is usually tackled by human operators using microscopes. Many industries, including medical and pharmaceutical, rely on the accuracy of this manual classification process, which is reported to be around 67%. In this paper, we propose a new method to automatically classify pollen grains using deep learning techniques that improve the correct classification rates in images not previously seen by the models. Our proposal manages to properly classify up to 98% of the examples from a dataset with 46 different classes of pollen grains, produced by the Classifynder classification system. This is an unprecedented result which surpasses all previous attempts both in accuracy and number and difficulty of taxa under consideration, which include types previously considered as indistinguishable.

## Introduction

Pollen is widely recognized as a nuisance, but also as a valuable tool in several scientific fields. An estimated 40% of the world's population experience seasonal allergic rhinitis (SAR) driven by exposure to pollen [1]. Pollen forecasting, informed by examination of airborne pollen, has become a key tool for management of SAR [2]. Pollen is also particularly important for quality verification of honey [3], reconstructing past vegetation to understand past changes in climate change [4], biodiversity [5], and human impacts [6] and as a forensic tool [7]. Common to all these areas is the need for experienced analysts to spend considerable amounts of time identifying and counting pollen on slides. While other branches of science have been transformed by the technological advances of recent decades, palynology is languishing, with the practical methodology of pollen counting having hardly advanced much beyond that of the 1950s.

But this is not for want of trying. Flenley [8] was the first to call attention to the need and potential of automation of pollen counting. A handful of early attempts were published in the later decades of the 20th century, but the rapid increase in capability in computational intelligence over the early part of the 21st century resulted in considerable acceleration in the field during this time, with numerous attempts at partial or complete automation of palynology appearing in the literature, summarized in [9] and [10].

**Competing interests:** The authors have declared that no competing interests exist.

While the results of the existing studies can be regarded as promising, they are rather limited in that they typically deal with a relatively small number of taxa (max 30, mean 8), and success/accuracy rates vary. While some palynological applications may require a lower level of taxonomic diversity than others, it is arguable that many 'real world' pollen problems will require higher diversity than that of most of the existing studies. For example, in [11], Stillman and Flenley, suggested that the minimum number of taxa for paleoecological applications would be around 40 types.

Recently, Sevillano and Aznarte in [10] presented an example of pollen classification which applied deep learning convolutional neural networks to the POLEN23E image dataset, a freely available set of 805 bright-field microscope images of 23 different pollen types from the Brazilian savannah [12]. Their model achieved accuracies of over 97%. 97% accuracy on a 23 class problem represents one of the best successes rate, when weighted for the number of taxa, of any attempt at automated pollen analysis currently documented in the literature.

In this paper, we apply the same approach to a wider set of pollen images produced by an automated pollen classification system, the Classifynder, marketed by Veritaxa ltd. This image set includes twice as many pollen types as the POLEN23E set, and over 19,000 individual images.

## State of the art

Previous approaches to automated palynology are comprehensively summarized in [9] and [10]. They can be divided into image-based and non-image based methods. Non-image based methods use alternative techniques for sensing characteristics used for differentiation, for example fluorescence [13], Fourier-Transform infrared [14], and Raman spectroscopy [15], etc. Non-image based methods won't be discussed further here.

Traditionally, image-based methods typically involve defining and extracting discriminating features from pollen images, followed by sorting via statistical or machine learning-based classifiers. An example which uses the same type of images as this research is [16]. As per [10], these image-based methods fall into three different categories based on the type of features being used: (1) discriminant features are largely visual/geometrical (e.g. shape, symmetry, diameter, etc.); (2) discriminant features are texture-based, i.e. they capture information about the pollen grain ornamentation (e.g. gray-level co-occurrence matrices,entropy features etc.,); and (3) a combination of the two approaches.

Recently, a new method for image-based pollen classification has emerged which harnesses the power of deep learning neural network frameworks. While still a combination of classification based on key image features, this new approach involves a model determining and extracting the features itself, rather than them being predefined by human analysts. Transfer learning is used to maximize classification ability by leveraging the representativeness of these synthetic features and the power of a linear discriminant classifier. In [17] a model learns not only the features but also the classifier itself from a deep learning neural network. The method makes use of transfer learning to leverage knowledge from pre-trained networks.

Existing studies applying this new approach outperform the traditional more supervised methods defined in 1-3 above. For example, [17] achieved a 94% of training accuracy on a dataset of 30 pollen types. Their technique is similar to ours, but their results are based on the training set and no information is given about how the model behaves with unseen images. The same can be said of [18], which informs of 100% accuracy on 10 very different pollen grains by using transfer learning over a convolutional neural network based on the VGG16 architecture, and of [19], which reports 99.8% training accuracy for 5 species. [10] experimented with three deep learning models for classification of pollen images from the

**Table 1. Previous attempts at automated classification of 10 or more pollen types, with number of taxa, classifier type and highest reported correct classification rate (CCR).** Abbreviations: NN = neural network, SVM = support vector machine, LDA = linear discriminant analysis, KNN = k-nearest neighbour, CNN = convolutional neural network, RNN = recurrent neural network.

|  | # taxa | classifier | reported CCR |
|---|---|---|---|
| Daood et al. (2016) [21] | 30 | CNN | 94% |
| Sevillano & Aznarte (2018) [10] | 23 | CNN | 97% |
| Goncalves et al. (2016) [12] | 23 | SVM | 64% |
| Kaya et al. (2013) [22] | 19 | Rough Set | 91% |
| Ticay-Rivas et al. (2011) [23] | 17 | NN | 96% |
| Marcos et al. (2015) [24] | 15 | KNN | 95% |
| Lagerstrom et al. (2015) [25] | 15 | LDA | 81% |
| Li et al. (2004) [26] | 13 | LDA, NN | 100% |
| Treloar et al. (2004) [27] | 12 | LDA | 100% |
| Khanzhina et al. (2018) [19] | 11 | CNN | 96% |
| Dhawale et al. (2013) [28] | 10 | NN | 85% |
| Daood et al. (2018) [18] | 10 | CNN+RNN | 100% |

POLEN23E dataset [12], and the same dataset has been recently used by [20]. These models use different combinations of automatic feature extraction, transfer learning and a pre-trained convolutional neural network.

Table 1 which provides a summary table of previous studies, including class sizes and accuracy/success rates.

## Materials and methods

### Pollen image set

The images used in our experiment are dark field microscope images captured on the Classifynder system (for an example, see Fig 1, for the full dataset see Supporting Information Section, in page 10). The Classifynder (formerly known as AutoStage) is documented in [29]. It was designed as a complete 'standalone' system for automated pollen analysis. The system uses basic shape features to identify the locations of objects of interest (i.e. pollen grains) in conventional microscope slides under a low-power objective. It then switches to a higher power objective and visits the location of each pollen object to capture an image of it to be used for classification. Objects are imaged at different focus levels, producing a 'Z-stack'. A 'Z-stack' is a set of images with different focus depths which show the different vertical details of a microscopic object. The system subsets the best-focused portions of each object Z-stack images, and then combines them to produce a single composite image, followed by segmentation from the background. This image is then falsely colored to show depth (Fig 1).

In its routine operation, the Classifynder extracts values for 43 different image features (both geometrical and textural, see [16] for a list) and tags them on to each image as metadata. Images are classified using a simple neural network (feed-forward with a single hidden layer), which compares the feature values of the unknown pollen types with the feature values of known 'library' images.

In this paper, we have used the Classifynder to generate our image set only, and have not used any of the Classifynder-generated image features or labeling. Our images were automatically collected from pollen reference slides. Images gathered from each reference slide were manually sorted, and were chosen with the intention of creating a set that was fully representative of the pollen type in question, i.e. accounting for variability in appearance resulting from

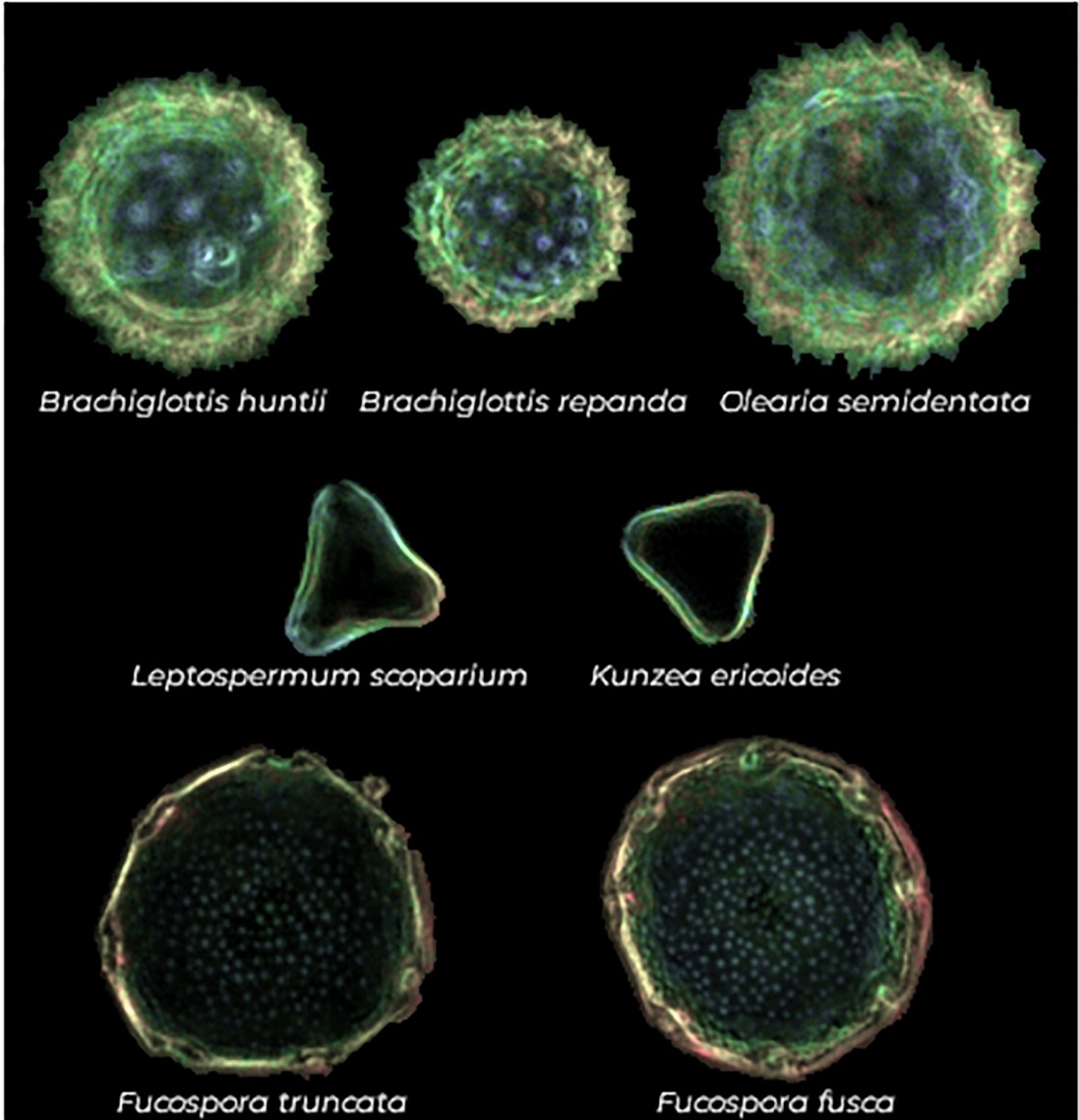

**Fig 1. Samples of seven pollen types found in the dataset.** By rows, pollen types usually considered as indistinguishable.

different viewing angles. Images with irregularities such as debris, or images containing multiple grains were excluded. While the Classifynder has produced satisfactory classification in previous experiments with 6 different types of taxa (Betula pendula, Dactylis glomerata, Cupressus macrocarpa, Ligustrum lucidum, Acacia dealbata and Pinus radiata) in [29], performance of the neural network classifier declines with 15 different types of taxa (Acacia Ramoissima, Atriplex Paludosa, Asteraceae, Casuarina Littoralis, Disphyma, Dracophyllum, Euphorbia Hirta, Eucalyptus Fasciculosa, Isoetes Pusilla, Myrsine, Nothofagus Cunninghamii, Nothofagus Discoidea, Nothofagus Discoidea, Olearia Algida, Phyllocladus, Aspleniifolius) in [16].

The original image dataset we use comprises a total of 19,500 images, from 46 different pollen types, representing 37 different families. This was the total number of taxa for which

suitable datasets were available at the time our research commenced. Data augmentation is applied on this original dataset.

Unlike the POLEN23E dataset, which is representative of the Brazilian savannah, our dataset is a mix of taxa found in New Zealand and the Pacific, including both native and introduced taxa. Many are types encountered in honeys from these regions.

The number of images per taxon varies from 40 to 1700. We have included the number of images per taxon below in Fig 8, in the Results and Discussion section (page 13). The majority of the images were captured from reference pollen slides, with the exception of 14 taxa (*Coprosma* sp., *Echium vulgare*, *Geniostoma* sp., *Griselinia* sp., *Ixerba brexiodes*, *Lotus* sp., *Lycopodium clavatum*, *Knightia excelsa*, *Metrosideros* sp., *Quintinnia* sp., *Ranunuculus* sp., *Salix* sp., *Taraxacum* sp., *Trifolium* sp., and *Weinmannia* sp.) whose images were captured from slides of pollen extracted from honey samples. All pollen samples, regardless of origin, were acetolysed following the method of Erdtman [30] and then suspended in either silicone oil or glycerine jelly, and mounted under cover-slips on glass slides.

Slides were scanned using the Classifynder system, which automatically locates and images the pollen grains, with only limited human input needed at the start of the process. It is this feature which has allowed for such a large image dataset to be generated, as we were not reliant on a human analyst to manually locate and image each grain.

The resultant raw image sets for each taxon or honey sample were manually examined. Image sets from honey samples were manually sorted into individual taxa, while image sets from reference slides were manually filtered to remove images of 'outliers', i.e. grains not representative of that pollen type due to deformation or malformation, as well as any non-pollen debris or pollen of other taxa that may have made their way onto the slide.

Overall, the number of images per taxon was dictated by what was available/gathered. As mentioned above, many of the individual sets were originally generated for other projects or experiments. This partially accounts for the considerable differences in the number of images per taxon. For example, the images for the taxa with the highest numbers of images (*Leptospermum scoparium* and *Kunzea ericoides*) were gathered as part of work on differentiating these two taxa using the Classifynder [31], and an even larger number of images were available for these two types. A figure of 1700 was arbitrarily selected.

The 46 taxa comprise a wide range of pollen morphologies, with some more similar than others. We have deliberately included two pairs and one trio of morphologically similar taxa to test the system's ability to discriminate closely related types, as well as across a broader range of morphologies. These are shown in Fig 1 and include 1) *Fuscospora fusca* and *F. truncata*, 2) *Leptospermum scoparium* and *Kuzea ericoides*, and 3) *Olearia semidentata*, *Brachyglottis huntii* and *B. repanda*. The last three are members of the *Asteraceae* family, many members of which are notoriously difficult to distinguish. *L. scoparium* and *K. ericoides* represent two important nectar sources in New Zealand honey, with *L. scoparium* being the source of the high-value New Zealand Mānuka honey. Traditionally, these two taxa have been regarded as virtually indistinguishable, limiting the use of pollen analysis in Mānuka honey quality assessment (although see [31] for more information). *F. fusca* and *F. truncata* are two species of southern beech (*Fagaceae*), whose pollen (along with the two other species of *Fuscospora*) are regarded as virtually identical [32].

## Deep learning convolutional neural network

Neural networks are inspired by the function and structure of the human brain. In these models, there are multiple layers of artificial neurons trained to process and identify concrete features of the input space, each layer extracting different valuable information. Deep learning is

a common name for the technique to train very complex neural networks that can be used on many types of data, like signal processing, image processing, speech recognition or natural language processing among many others, to produce results that often are similar to those that a human being would produce.

In the field of image recognition, deep learning of neural networks has reached levels of accuracy not previously achieved. While traditional neural networks contain a few hidden layers of neurons, deep learning networks can contain tens or hundreds. These models are trained with large data sets and can learn features without the need for manual intervention. This ability to extract features from large data sets makes them especially suited for the task of the classification of pollen grains, where the correct identification of these features is especially complex. Eliminating the manual selection of features significantly simplifies the classification process, while increasing the reliability.

There are different techniques to create and train these models. The most common three are 1) training from scratch, where the network is built from the beginning, 2) transfer learning, where the structure of a pre-trained model is adapted and 3) feature extraction, a more specialized approach in which the learned features are used as input for another automatic learning model as, for example, a support vector machine or a linear discriminant classifier. In this work, we present a hybrid solution between the last two configurations.

First, we use the pre-trained network Alexnet [33] to automatically extract significant features of the images. Alexnet is one of the most popular convolutional neural networks, and it was originally designed to classify images from 1000 different classes. It is much larger than previous convolutional neural networks, consisting of 5 convolutional layers and 3 fully connected layers, and having around 60 million parameters and 650,000 neurons.

Through the different layers we can observe what features the network learns by comparing areas of activation with the original image. The first layers learn basic features like color and edges, while in the deeper layers the network learns more complex and abstract features. Identifying features in different layers allows us to understand what the network has learned. In Fig 2 we show an example of nine of the trained filters for each of the five convolutional layers of the network. These filters allow deep layers to detect more complex patterns. The deeper layers can learn higher level combinations of features learned by the previous layers. After these layers, Alexnet contains three fully connected layers, of which the last contains 1000 neurons, one for each class. The $i$th neuron of this layer is interpreted as the probability that the input image belongs to the $i$th class. In our model, we extract the automatically created features from the second fully connected layer because it is the deeper layer and it is thus expected to contain the higher level representation of the differences amongst the classes.

The features extracted by these layers usually do not offer an overly clear vision of the network operation. We can, however, analyze the activations that a concrete image produces when processed through the network to obtain its classification. In this case, the different

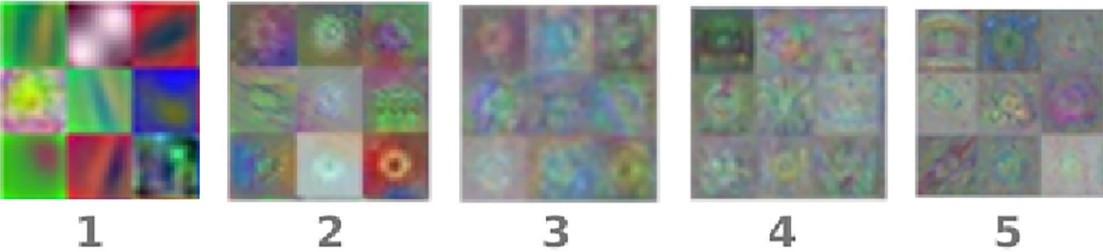

**Fig 2. Example of 9 filters from the five convolutional layers of the trained network.**

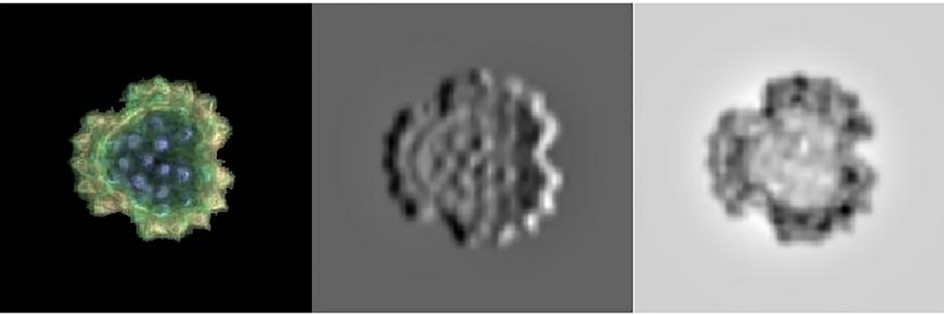

**Fig 3. A sample *Brachyglottis huntii* image (left) and the activation it produced in filters 29 (center) and 85 (right) of the first layer of the convolutional neural network.**

filters and activations in the different layers are clearer to the naked eye. To illustrate the example, an image belonging to the *Brachyglottis huntii* class has been selected (see Fig 3).

The image is initially preprocessed to be adapted to the network requirements as described below. We then feed the image to the trained convolutional neural network and display the activations of different layers of the network to, later, examine these activations and analyze what features the network learns by comparing them with the original image. Fig 3 shows the activations that the image produces in two of the filters of the first convolutional layer. White pixels represent strong positive activations and black pixels represent strong negative activations. When a filter is mostly gray, it does not respond to any strong activation on the original image. The position of the pixels on the activations corresponds to the same position on the original image. This way, we can analyze which part of the image is identified by each of the activated filters.

For filter 29 (center of Fig 3) we can see positive and negative activations, in black and white respectively, corresponding to the features detected by this filter. The positive activations of this filter recognize the edges, identifying the external peaks of the image and the internal purple points. Filter 85, in the right hand side of Fig 3, is the one showing lager activations for the first convolutional layer and the selected sample image.

The deeper layers contain filters that have learned to identify more complex features from the previous layers. A closer look at all the activations produced by this sample image in the convolutional layer 5, in Fig 4, bring us closer to the complexity of the features learned by this deep layer. Considering the whiter areas as positive activations, we can observe, for example, how the layer learns about the external shape of the grain.

As mentioned above, after this layer another three fully connected layers are present, the last layer being an output layer. Once the network is trained, we extract the relevant features from the second fully connected layer and we train a linear discriminant classifier to perform the classification on these features.

## Experimental design

To properly estimate the error and to assess the robustness of the models against overfitting, the accuracy on each conducted experiment was computed using a 10 fold cross-validation process summarized in Fig 5. As we can see in this figure, the images are divided into two subsets, training and validation, comprising 90% and 10% of the images respectively. This division occurs ten times, or folds, guaranteeing that all the images have been present at least once in both sets, training and validation. The data augmentation process is performed on the training set, once the validation images have been separated, to avoid bias in the model.

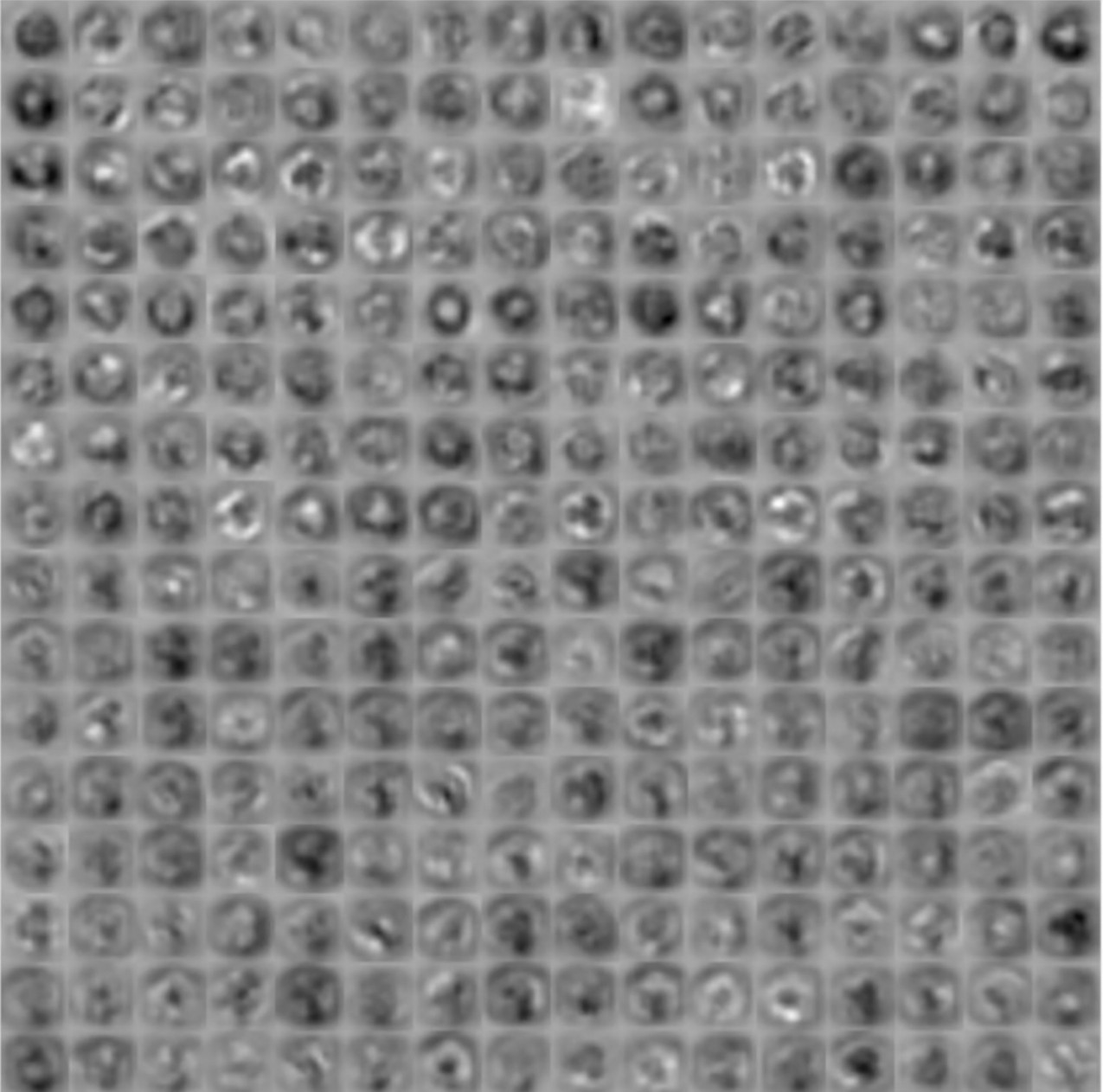

**Fig 4. Fifth convolutional layer activations for the *Brachyglottis huntii* class sample image of Fig 3.**

We compute the following common performance measures to evaluate model performance:

$$CCR = \frac{TP + TN}{TP + TN + FP + FN} \tag{1}$$

$$precision = \frac{TP}{TP + FP} \tag{2}$$

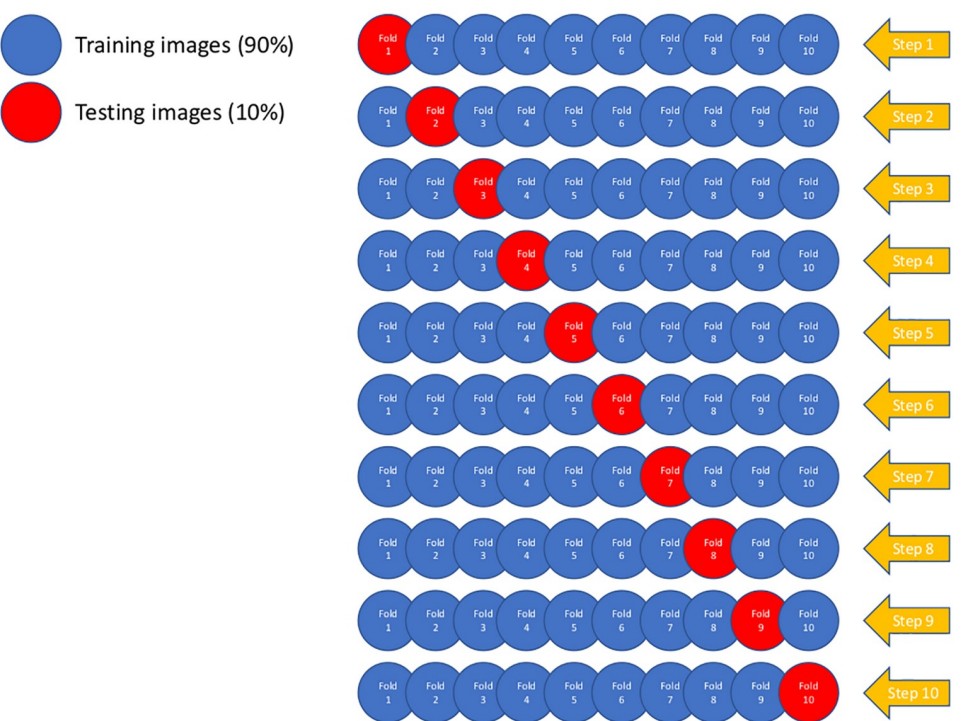

**Fig 5. Diagram for the cross-validation procedure.** The red circles represent the validation sets for each step, containing 10% of the images for each class.

$$\text{recall} = \frac{TP}{TP + FN} \quad (3)$$

$$\text{F1 score} = 2 * \frac{\text{precision} * \text{recall}}{\text{precision} + \text{recall}} \quad (4)$$

where CCR indicates the correct classification rate, *TP* refers to true positives, *TN* to true negatives, *FP* to false positives, and *FN* to false negatives. Precision, recall and F1 score were computed as an average weighted by the number of images in each class.

**Image preprocessing and augmentation.** In our experiment, in addition to performing a custom image preprocessing procedure, we implemented data augmentation, which considerably increases the convolutional neural network training time, and the training time of the linear discriminant classifier, but that also increases the accuracy accordingly. This process is summarized in Fig 6.

One of the most relevant features of the dataset is the image size. This feature is important since different pollen grains of different classes have different sizes, and thus this feature is essential for classification. As the convolutional neural network needs a 227x227x3 format as an input, all images must be re framed into that format. Our preprocessing algorithm crops the grains from the original images while keeping a minimum padding around them and maintaining their size so that the network can identify it more effectively. This reduction of the images to their effective content facilitates the process of data augmentation, since it allows to generate new images from the originals without resizing them in the process. This technique allows to balance the amount of images among different classes but also provides a wider variety of spatial features. By rotating the image, relevant features can be identified in different

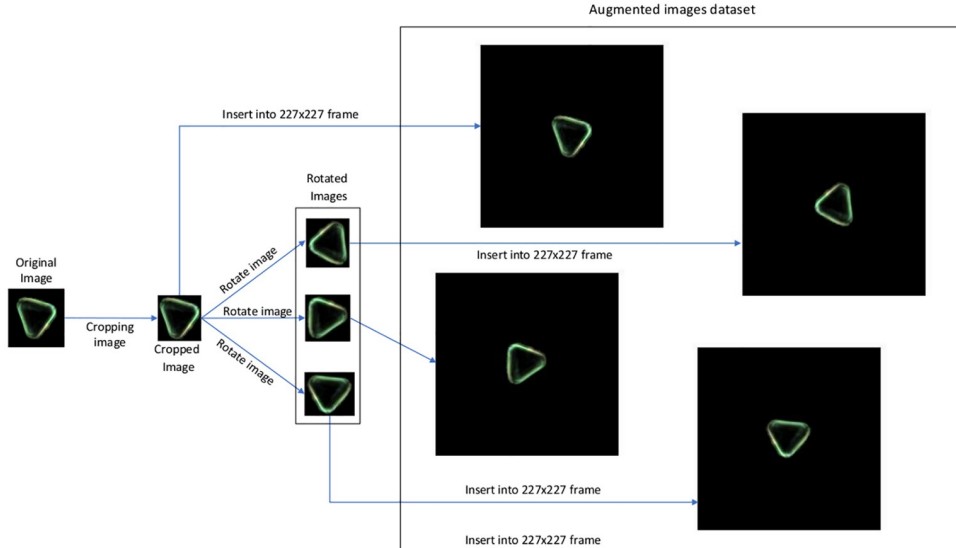

**Fig 6. Summary of the data augmentation process applied.**

locations, helping to provide spatial invariance. The result is equivalent to obtain samples in different orientations from the microscope. After that, we insert the cropped and rotated image in a 227x227 black frame.

In the framework of convolutional neural networks it is common to derive an augmented image dataset by generating batches of new images, with optional preprocessing such as resizing, rotation, and reflection, based on the available images. Augmenting image data helps to prevent the network from over-fitting and memorizing the exact details of the training images. It also increases the effective size of the training data set.

Furthermore, the original set of images from this experiment has classes with very different amounts of images, ranging from 1700 in the most represented case to 40 in the least. In order to obtain the greatest number of features of each class while maintaining a reasonable training time, we have increased the images of each class to the number of images of the class with more examples.

Hence, from the original preprocessed images, subsequent synthetic images are obtained to complete the image data set. The creation process we have used relies only on the rotation of images. A resizing process was not considered to be effective since they would lose the aforementioned size ratio. Each new image is generated from a random image of its class. Once the original image has been randomly selected, a rotation angle, also randomly chosen, is applied, generating an rotated image from the original one, but maintaining the dimensions of it, even if the final image contains a different canvas size.

This technique allows to generate synthetic images that preserve the original proportions but contain different characteristics, significantly increasing the features presented to the convolutional neuronal network.

## Results and discussion

As explained above, to properly evaluate the model performance, we use a validation set composed of images not seen by the model during the training process. These images constitute an

**Table 2. Results for the training and validation sets.**

|  | CCR | Precision | Recall | F1 score |
|---|---|---|---|---|
| Training | 99.91% | 0.998 | 1.000 | 0.999 |
|  | (± 0.011) | (± 0.021) | (± 4.84e-04) | (± 0.011) |
| Validation | 97.86% | 0.979 | 0.978 | 0.978 |
|  | (± 0.252) | (± 0.030) | (± 0.031) | (± 0.027) |

example close to reality, since they are completely unknown to the model and allow us to anticipate its behavior against new observations.

Table 2 presents the results obtained by the model using the training and the validation datasets, as an average of the 10 cross-validation partitions together with the standard deviation of each value, in brackets. The CCR obtained by the examples of the validation set rises to 97.86%, which approximates the results obtained during training, 99.91%.

The low deviation between training and validation values demonstrates the model robustness, ruling out the possibility of overfitting during the training process. This is supported by the low values in the standard deviations of the different measures, which also prove the stability of the model.

However, the CCR measure alone does not provide sufficient information about the model behavior against false positives and false negatives produced in the process of predicting new observations. The values obtained for precision and recall, 0.979 and 0.978 respectively reinforce the consistency of the model with respect to false positives and false negatives. Finally, the high value of the F1 score validates the consistency of those measures.

In order to have the full picture and shed light into how the model behaves with different classes, Fig 7 shows the confusion matrix obtained by the model over the 10 validation datasets of the cross-validation procedure. This matrix has been constructed by accumulating the individual cross-validation results: since each validation set was composed by a 10% of the data for each type, and we had 10 such datasets, the matrix columnwise sums are equal to the total of images for each type. We can dive into this matrix by looking at the aforementioned 3 sets of virtually indistinguishable pollen types.

With regards to *Olearia semidentata* and *Brachyglottis repanda*, we see how the model wrongly predicts as *O. semidentata* 29 images that should have been classified as corresponding to *B. huntii*. This represents a 9% of the total for the latter. Conversely, 16 images (8%) of *O. semidentata* are wrongly classified as *B. huntii*. These comprise every misclassification recorded for both classes. Furthermore, there was only one confusion between *B. repanda* and *B. huntii*.

When dealing with *Fuscospora fusca* and *F. truncata*, our model gets 16 images of the former misclassified as the latter, representing a 4.8% of the total. And conversely, 12 images of *F. truncata* are misclassified as *Fuscospora fusca* (6.7%).

Finally, concerning *Leptospermum scoparium* and *Kunzea ericoides*, we see that 42 images of the latter (representing only a 2.5%) are misclassified as *Leptospermum scoparium*, whereas 73 (4.3%) of the former are also misclassified as the other. Given the known similarities amongst both types, these results can be considered as very good (and are in fact far better than what a human operator can achieve).

Fig 8 shows a boxplot with the distribution of the F1 scores for the validation sets during the cross-validation process. In this figure, we can observe the good general behavior of the model, with two types perfectly classified in all the cross-validation partitions (*Carpodetus* and *Gunnera*), with other 10 types perfectly classified in median (*B. repanda*, *Carex*, *Geniostoma*,

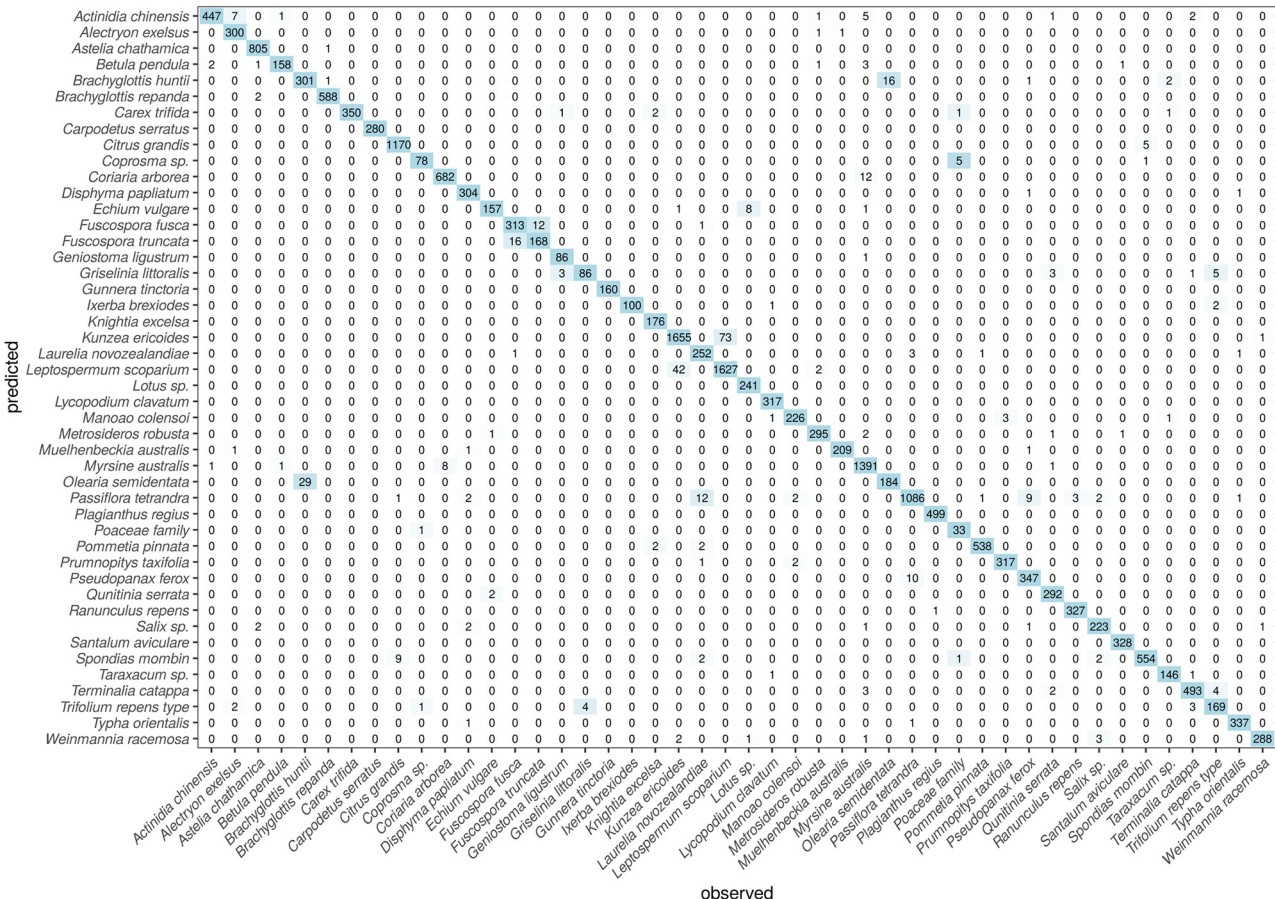

**Fig 7. Confusion matrix for the 46 pollen types through the 10-fold cross validation process.** For each type, 10% of the available images were saved for validation.

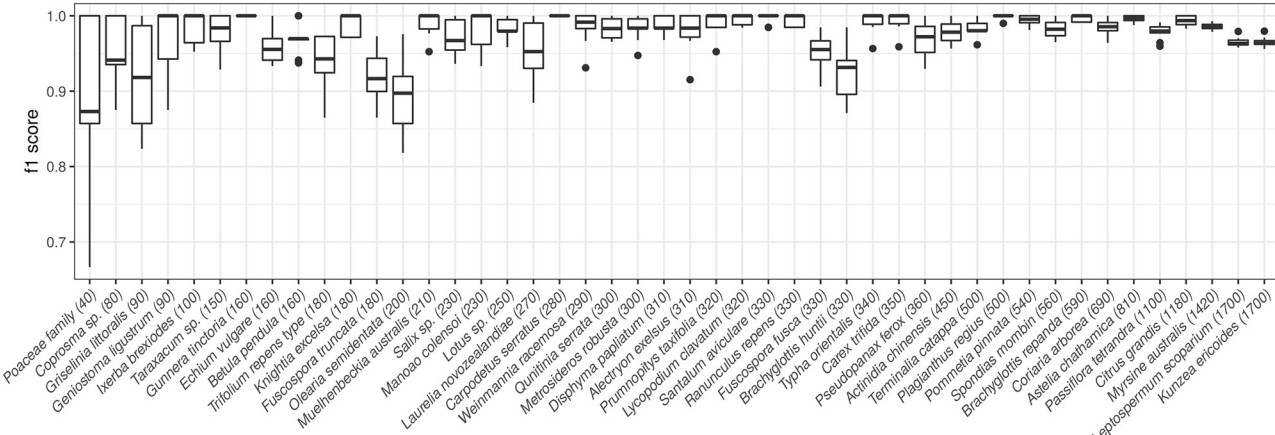

**Fig 8. Distribution of f1 score per type in the 10 folds of the cross-validation process.** Taxa are ordered with respect to the number of images for each type, which is included in brackets next to the taxon name.

*Ixerba*, *Knightia*, *Lycopodium*, *Maonao*, *Muelhenbeckia*, *Plagianthus*, *Prumnopitys*, *Ranunuculus*, *Santalum* and *Typha*), and with medians above 0.95 for all but 7 of the classes.

In Fig 8, the 46 pollen types are ordered according to the number of images available for each one. This reveals a pattern: leaving aside the indistinguishable pairs mentioned above, which of course present lower F1 than others, it seems that there is a relationship between the number of available images and the results. As expected, when the number of images increases, the error of the model decreases: all the types that obtain a median F1 below 0.95 are either part of the indistinguishable pairs or have less than 200 images.

This is evident in the case of the *Poaceae* class, which is the one with the fewest original images, having only 40 in total (and thus only 4 in the validation sets). This of course implies a greater dependence on the selection of samples in the different cross-validation experiments. It is also clear that the difficulties with this particular class are responsible for a significant decrease in the model overall accuracy.

## Conclusions

Finally, we can add some concluding remarks. In this work we present a model for the classification of images of 46 different kinds of pollen grains captured with the Clasifynder automatic microscope. We have used different techniques of image pre-processing and data augmentation to feed a pre-trained convolutional neural network, retrained by transfer learning to extract features from one of its deepest layers. Finally, these automatically extracted features are used to perform classification with a linear discriminant classifier.

The behavior of the model is excellent, with an accuracy higher than 97% in unseen sets of images. Furthermore, we have proven how it is able to correctly set apart pairs of pollen types considered indistinguishable by palynologists. The performance was slightly lower for those types with less images available, pointing at even higher overall accuracies using vaster, more complete datasets.

These groundbreaking results are of great interest for the automation of pollen counting, identified as one of the future achievements in the palynology field.

## Supporting information

**S1 File.**
(TXT)

## Author Contributions

**Conceptualization:** Katherine Holt, José L. Aznarte.

**Data curation:** Víctor Sevillano, Katherine Holt, José L. Aznarte.

**Formal analysis:** Katherine Holt, José L. Aznarte.

**Funding acquisition:** Katherine Holt, José L. Aznarte.

**Investigation:** Víctor Sevillano, Katherine Holt, José L. Aznarte.

**Methodology:** José L. Aznarte.

**Project administration:** José L. Aznarte.

**Resources:** Katherine Holt, José L. Aznarte.

**Software:** Víctor Sevillano.

**Supervision:** José L. Aznarte.

**Validation:** Katherine Holt, José L. Aznarte.

**Visualization:** Víctor Sevillano, José L. Aznarte.

**Writing – original draft:** Víctor Sevillano, Katherine Holt, José L. Aznarte.

**Writing – review & editing:** Katherine Holt, José L. Aznarte.

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
