## [Decision Letter · Decision Letter 0]

10 Apr 2020

PONE-D-20-04167

Precise automatic classification of 46 different pollen types with convolutional neural networks

PLOS ONE

Dear Dr. Aznarte,

Thank you for submitting your manuscript to PLOS ONE. After careful consideration, we feel that it has merit but does not fully meet PLOS ONE’s publication criteria as it currently stands. Therefore, we invite you to submit a revised version of the manuscript that addresses the points raised during the review process.

We would appreciate receiving your revised manuscript by May 25 2020 11:59PM. To enhance the reproducibility of your results, we recommend that if applicable you deposit your laboratory protocols in protocols.io, where a protocol can be assigned its own identifier (DOI) such that it can be cited independently in the future. For instructions see: http://journals.plos.org/plosone/s/submission-guidelines#loc-laboratory-protocols

We look forward to receiving your revised manuscript.

Kind regards,

Mudassar Raza, Ph.D.

Academic Editor

PLOS ONE

Journal Requirements:

Reviewers' comments:

Reviewer's Responses to Questions

**Comments to the Author**

1. Is the manuscript technically sound, and do the data support the conclusions?

Reviewer #1: Yes

Reviewer #2: No

Reviewer #3: Yes

2. Has the statistical analysis been performed appropriately and rigorously? 

Reviewer #1: Yes

Reviewer #2: N/A

Reviewer #3: Yes

3. Have the authors made all data underlying the findings in their manuscript fully available?

Reviewer #1: Yes

Reviewer #2: Yes

Reviewer #3: Yes

4. Is the manuscript presented in an intelligible fashion and written in standard English?

Reviewer #1: No

Reviewer #2: Yes

Reviewer #3: Yes

5. Review Comments to the Author

Reviewer #1: The manuscript presents the computational experiment regarding highly accurate classification of the pollen samples based on their microscopic images. The described methodology has gradually emerged from the authors’ previous works which is clearly described and referenced. All expected elements of explanation are provided, in particular: the motivation, new details as compared to the previous works, comparative features when confronted with state-of-the-art references, etc. There are two main areas referred to in the manuscript: a) biological data – i.e. pollen images – along with the need to recognize the type from the image, b) convolutional neural network computational tool. Both are introduced and described in the manner that seems clear enough to the other branch of readers. Some tutorial elements have been provided and the details that were somewhat out of scope have been properly referenced. The results are presented in the carefully designed manner – informative however not overdone. The overall impression is that even particularly keen readers looking for some specific details should be satisfied. In particular, I approve the presented computational and experimental methodology.

I have marked ‘no’ in the category of language quality but only because there were two options (yes/no), as there are only some few typos which need correction, in the manuscript – for example, in the Results and discussion section, right below Table 1: ‘The’.

1. In the Introduction section: some regular references, like [x], would be expected where only the year, like (1996) or (2016), is mentioned.

2. In Materials and Methods/Experimental design: CCR abrv. is not explained, while TP, TN, etc., are – and they all seem to belong to similar level of common knowledge.

3. I wonder if grouping the hard-to-distinguish taxa into ‘grouped categories’ could make any sense from the pollen classification point of view, but then the data presented in Table 1 might look even more promising – in the form of new Table 2 (The results in Table 1 should be kept). It seems that with the resulting data, already at hand, one could easily calculate all the quality assessing parameters for just 42 categories, when three of them would be composed of more than one taxon. The results of kind as in Fig. 8 could be also added for such case.

4. The Acknowledgements section is in Latin, which a bit strange and, unfortunately, unreadable to me.

Reviewer #2: In the study, the authors propose using convolutional neural networks to classify pollen images produced by the Classifynder classification system and claim their proposed method outperform all previous attempts both in accuracy, class number and indistinguishable classes. The manuscript is clear and easy to follow.

Though it has some merits, it suffers some severe problems, which unfortunately cannot be remedied from my point of view.

First of all, they ignore the fact that the Classifynder is embedded with a neural network used to classify pollen images. Since the class labels generated by this neural network is used as the ground truth, no other classification algorithms (including deep learning) can beat this embedded algorithm because its CCP, precision, recall and F1 score are always 100%. Unless there are other pollen image data sets labeled in other approaches (for example manually labeled pollen images), I don’t think they can demonstrate that their proposed method surpasses previous attempts.

Second, a method is claimed to be better than previous attempts on classification but without any quantitative comparison in its result and discussion, which in principle is faulty and not credible. In addition, their results are redundant. The Table 1 shows that the proposed method can achieve performances (CCR, precision, recall and F1 score) greater than 97%. However, they virtually repeat this information by digging into confusion matrix because CCR, precision, recall and F1 score are calculated summaries based on the confusion matrix. The confusion matrix can’t add more information from a different angle than the Table 1. Neither F1 score by type.

At last, the manuscript spends a large space to explain some basic machine learning concepts, such as 10-fold cross validation (Fig. 5) and performance measures etc. It gives me the impression of stuffing the manuscript with unnecessary content.

Overall, I don’t think this manuscript can be improved significantly, especially if there are no other pollen images with labels.

Reviewer #3: In this paper Sevillano et al. use convolutional neural networks for automatic classification of 46 pollen types. Deep learning methods improve the accuracy of the classification over manual methods from 67% to 98%. The authors apply the methodology used on a previous report (their reference [10]) to a larger data set that included a larger number of pollen types. They use a large number of images (19,500) corresponding the 46 pollen types. The number of images per taxon ranges from 40 to 1,700.

The authors use a standard pre-trained convolutional neural network (Alexnet) and a conventional experimental design to train their neural network and analyze its performance. The study shows an impressive increase in classification performance when using deep learning methodologies. It would be helpful that the authors also present a table comparing the performance of other published pollen classifiers including the number of images and pollen types used on those studies.

Below follow some comments and questions that may improve the readability and impact of this paper.

1) The authors should provide a reference for the method in paragraph between lines 54-60.

2) Please describe briefly what is a Z-stack (line 82).

3) To improve readability of the paper, perhaps you can provide the accuracy rate for low and high number of taxa of the Classifynder neural network. (line 93).

4) Figure 8 is discussed in the text after Figure 1. Should then Figure 8 be Figure 2?

5) It is not clear what the reader can learn from Figure 2 (line 183). Perhaps the authors could explain with a greater depth what is in the figure.

6) The authors assume that layer 5 (Figure 4) learns about complex characteristics of the pollen type. Can the authors provide some insight on what these characteristics could be from the activations shown in Figure 4?

7) Does the number of images used (19,500) include those produced by data augmentation, or 19,500 is the number of original images?

8) Can you clarify how the rotation of the original figures increases the features presented to the neural network? Which is the value added of rotating an image? Wouldn’t rotation just create redundant data?

9) How were the training and validation sets constructed? Since the classification performance depends on the number of images available (and therefore on the pollen type, Figure 8), the overall performance reported in Table 1 may depend critically on how the sets were selected.

6. PLOS authors have the option to publish the peer review history of their article (what does this mean?). If published, this will include your full peer review and any attached files.

Reviewer #1: No

Reviewer #2: No

Reviewer #3: No

---

## [Author Response · Author response to Decision Letter 0]

6 May 2020

Response to Reviewer #1

Comment 0: I have marked ‘no’ in the category of language quality but only because there were two options (yes/no), as there are only some few typos which need correction, in the manuscript – for example, in the Results and discussion section, right below Table 1: ‘The’.

Response: We thank the reviewer for his/her call, and we have reviewed the grammar.

Comment 1: In the Introduction section: some regular references, like [x], would be expected where only the year, like (1996) or (2016), is mentioned.

Response: References and citations have been reviewed.

Comment 2: In Materials and Methods/Experimental design: CCR abrv. is not explained, while TP, TN, etc., are – and they all seem to belong to similar level of common knowledge

Response: The definition for CCR (Correct Classification Rate) has been included in the “Experimental design” section.

Comment 3: I wonder if grouping the hard-to-distinguish taxa into ‘grouped categories’ could make any sense from the pollen classification point of view, but then the data presented in Table 1 might look even more promising – in the form of new Table 2 (The results in Table 1 should be kept). It seems that with the resulting data, already at hand, one could easily calculate all the quality assessing parameters for just 42 categories, when three of them would be composed of more than one taxon.

The results of kind as in Fig. 8 could be also added for such case.

Response: As part of our future work, we are considering the probabilities of each taxa in reference to each of the different classes. In this way, for those classes that have similar probabilities, we will be able to indicate what classes they resemble instead of giving an absolute classification.

Comment 4: The Acknowledgements section is in Latin, which a bit strange and, unfortunately, unreadable to me.

Response: In reference to this point, a section of the document template was unintentionally included. We have proceeded to eliminate it. We apologize for the inconvenience. 

Response to Reviewer #2

Comment 1: First of all, they ignore the fact that the Classifynder is embedded with a neural network used to classify pollen images. Since the class labels generated by this neural network is used as the ground truth, no other classification algorithms (including deep learning) can beat this embedded algorithm because its CCP, precision, recall and F1 score are always 100%. Unless there are other pollen image data sets labelled in other approaches (for example manually labelled pollen images), I don’t think they can demonstrate that their proposed method surpasses previous attempts.

Response: Reviewer 2 appears to have assumed that we have used the native neural network of the Classifynder to provide the initial identifications of the images. This is incorrect. We have not used the neural network of the Classifynder in any part of the research presented here, and this is made clear in the statement within the methods "In this paper, we hav used the Classifynder to generate our image set only."

Furthermore, we cite papers that clearly demonstrate that the native neural network simply cannot produce similar levels of accuracy in classification as the approach used here, i.e. "While the Classifynder has produced satisfactory classification in previous experiments with low numbers of taxa (e.g. Holt et al. 2011), performance of the neural network classifier declines with greater numbers of taxa (e.g. Lagerstrom et al. 2015)". This later paper demonstrates that precision from the native neural network is far from 100%.

This has been clarified in the Material & Methods section of the paper with the following text: "Images gathered from each reference slide were manually sorted, and were chosen with the intention of creating a set that was fully representative of the pollen type in question, i.e. accounting for variability in appearance resulting from different viewing angles. Images with irregularities such as debris, or images containing multiple grains were excluded".

Comment 2: Second, a method is claimed to be better than previous attempts on classification but without any quantitative comparison in its result and discussion, which in principle is faulty and not credible. In addition, their results are redundant. The Table 1 shows that the proposed method can achieve performances (CCR, precision, recall and F1 score) greater than 97%. However, they virtually repeat this information by digging into confusion matrix because CCR, precision, recall and F1 score are calculated summaries based on the confusion matrix. The confusion matrix can’t add more information from a different angle than the Table 1. Neither F1 score by type.

Response: In the first version of the paper, we did not provide a detailed comparison with previous attempts in the discussion. We provided a summary of previous attempts in the 'State of the Art' section, which highlights some of the highest rates of accuracy. But we also made the point that the majority of these previous studies are working with smaller number of classes, so even if they achieve equal or higher precision, they are arguably less valuable in a practical sense, as they are less likely to be useful in solving real world pollen classification problems. We didn't think there was much point in performing a detailed quantitative comparison in the discussion, as we are not able to compare

'apples with apples', in that there are no other studies that have examined a pollen Classification problem on the same scale as this. However, we have added a new table to the paper, based on the one by Holt and Bennett (2014), which provides a summary table of previous studies, including class sizes and accuracy/success rates.

With regard to confusion matrix and F1 score by type, we must disagree with Reviewer 2 since both figures do add information with respect to Table 3 (former Table 1). For example, it is possible to determine those pollen types which are difficult and those easy to distinguish, something that is impossible if we stick to Table 3 (former Table 1) alone.

Comment 3: At last, the manuscript spends a large space to explain some basic machine learning concepts, such as 10-fold cross validation (Fig. 5) and performance measures etc. It gives me the impression of stuffing the manuscript with unnecessary content.

Response: Since the paper is aimed at a broader audience than machine learning experts, we consider necessary to clarify some of the main concepts used in our work. An example is cross-validation: as the review of the state of the art suggests, not all previous attempts were aware of the available error estimation procedures, and their main drawbacks. We thus must insist in leaving that section as it is, despite the "stuffing the manuscript" indication by Reviewer 2.

Comment 4: Overall, I don’t think this manuscript can be improved significantly, especially if there are no other pollen images with labels.

Response: Given the fact that the main objection of Reviewer 2 is not sustained, we hope he or she will find the paper interesting.

Response to Reviewer #3

Comment 0: It would be helpful that the authors also present a table comparing the performance of other published pollen classifiers including the number of images and pollen types used on those studies.

Response: Thanks to Reviewer 3 for his/her indications. We have added a new table (Table 1) to the paper with a summary of the results of previous publications which are comparable with ours.

Comment 1: The authors should provide a reference for the method in paragraph between lines 54-60.

Response: We added a reference for the method in the “State of the art” section.

Comment 2: Please describe briefly what is a Z-stack (line 82).

Response: We have included the definition in the article as follows: “A Z-stack is a set of images with different focus depths which show the different vertical details of a microscopic object.”

Comment 3: To improve readability of the paper, perhaps you can provide the accuracy rate for low and high number of taxa of the Classifynder neural network. (line 93). 

Response: We modified the paragraph as follows: “While the Classifynder has produced satisfactory classification in previous experiments with 6 different types of taxa (Betula pendula, Dactylis glomerata, Cupressus macrocarpa, Ligustrum lucidum, Acacia dealbata and Pinus radiata) in [21], performance of the neural network classifier declines with 15 different types of taxa (Acacia Ramoissima, Atriplex Paludosa, Asteraceae, Casuarina Littoralis, Disphyma, Dracophyllum, Euphorbia Hirta, Eucalyptus Fasciculosa, Isoetes Pusilla, Myrsine, Nothofagus Cunninghamii, Nothofagus Discoidea, Nothofagus Discoidea, Olearia Algida, Phyllocladus, Aspleniifolius) in [16]”.

Comment 4: Figure 8 is discussed in the text after Figure 1. Should then Figure 8 be Figure 2?

Response: Indeed we have a problem with this, as the data corresponding to the number of images per taxa is included in Figure 8 (and implictly in Figure 7), both in the Results section, but mentioned earlier on in the text. One alternative is to include in the Material and Methods a new table with the number of images per taxa, but we believe this will take an amount of space which is not justified by the interest. Thus, we decided to maintain the reference, explaining a bit more in the paper the reason for this inconsistency and pointing to the page in which the Figure 8 is displayed. If the editor or the reviewer consider that including this new table is a better solution, we can easily do so (it’s already coded in the source files).

Comment 5: It is not clear what the reader can learn from Figure 2 (line 183). Perhaps the authors could explain with a greater depth what is in the figure.

Response: We added the following explanation for these filters in the Deep learning convolutional neural network section: “These filters allow deep layers to detect more complex patterns. The deeper layers can learn higher level combinations of features learned by the previous layers.”

Comment 6: The authors assume that layer 5 (Figure 4) learns about complex characteristics of the pollen type. Can the authors provide some insight on what these characteristics could be from the activations shown in Figure 4?

Response: We added an example of what the layer learns from positive activations. “Considering the whiter areas as positive activations, we can observe, for example, how the layer learns about the external shape of the grain.”

Comment 7: Does the number of images used (19,500) include those produced by data augmentation, or 19,500 is the number of original images? 

Response: The 19,500 images conform the original image data set from where we apply data augmentation techniques as described in “Materials and methods”, section “Pollen image set” as follows: “The original image dataset we use comprises a total of 19,500 images, from 46 different

pollen types, representing 37 different families. This was the total number of taxa for which suitable Classifynder datasets were available at the time our research commenced. Data augmentation is applied on this original dataset.” 

Comment 8: Can you clarify how the rotation of the original figures increases the features presented to the neural network? Which is the value added of rotating an image? Wouldn’t rotation just create redundant data?

Response: Rotating the images is a quite common technique in computer vision to balance data sets with hugely different number of images, and at the same time, helping to identify features in distinct locations.

We added the following text in the Image preprocessing and augmentation section to clarify it: “This technique allows not only to balance the number of images among different classes but also provides a wider spatial features variety. By rotating the image, relevant features can be identified in different locations, helping to provide spatial invariance. The result is equivalent to obtain samples in different orientations from the microscope.”

Comment 9: How were the training and validation sets constructed? Since the classification performance depends on the number of images available (and therefore, on the pollen type, Figure 8), the overall performance reported in Table 1 may depend critically on how the sets were selected.

Response: We have included the following text in the “Experimental design” section to explain 

the cross-validation process: “As we can see in this figure, the images are divided into two subsets, training and validation, at 90% and 10% respectively. This division occurs ten times, folds, guaranteeing that all the images have been present in both sets, training and validation. The data augmentation process is performed on the training set, once the validation images have been separated, to avoid bias in the model.”

---

## [Decision Letter · Decision Letter 1]

20 May 2020

Precise automatic classification of 46 different pollen types with convolutional neural networks

PONE-D-20-04167R1

Dear Dr. Aznarte,

We are pleased to inform you that your manuscript has been judged scientifically suitable for publication and will be formally accepted for publication once it complies with all outstanding technical requirements.

With kind regards,

Mudassar Raza, Ph.D.

Academic Editor

PLOS ONE

Reviewers' comments:

Reviewer's Responses to Questions

**Comments to the Author**

1. If the authors have adequately addressed your comments raised in a previous round of review and you feel that this manuscript is now acceptable for publication, you may indicate that here to bypass the “Comments to the Author” section, enter your conflict of interest statement in the “Confidential to Editor” section, and submit your "Accept" recommendation.

Reviewer #1: All comments have been addressed

Reviewer #3: All comments have been addressed

2. Is the manuscript technically sound, and do the data support the conclusions?

Reviewer #1: Yes

Reviewer #3: Yes

3. Has the statistical analysis been performed appropriately and rigorously? 

Reviewer #1: Yes

Reviewer #3: Yes

4. Have the authors made all data underlying the findings in their manuscript fully available?

Reviewer #1: Yes

Reviewer #3: Yes

5. Is the manuscript presented in an intelligible fashion and written in standard English?

Reviewer #1: Yes

Reviewer #3: Yes

6. Review Comments to the Author

Reviewer #1: I definitely accept the authors response and the changes they have introduced, accordingly.

I have only one subtle comment, which I think should be left to the authors to decide about.

In Fig. 7 the graphical summaries of resulting statistical distributions are presented and it seems to be one of the most vital parts of the material. Therefore, it appears to be some waste of space when only the first plot decides about the range (along the f1 axis) of the whole graph. Consequently, some of the plots lose their clarity or are just unreadable. Could the authors think of some way of conveying the message that the whisker of the first plot sticks out beyond the graph and simultaneously adapt the range so that it covers the remaining plots only – obviously resulting in the relevant extension of all box plots.

I do not have more comments and regard the manuscript certainly good enough to be accepted.

Reviewer #3: (No Response)

7. PLOS authors have the option to publish the peer review history of their article (what does this mean?). If published, this will include your full peer review and any attached files.

Reviewer #1: Yes: Przemysław Korohoda, AGH University of Science and Technology, Kraków, Poland

Reviewer #3: No

---

## [Editor Report · Acceptance letter]

10 Jun 2020

PONE-D-20-04167R1 

Precise automatic classification of 46 different pollen types with convolutional neural networks 

Dear Dr. Aznarte:

I'm pleased to inform you that your manuscript has been deemed suitable for publication in PLOS ONE. Congratulations! Your manuscript is now with our production department. 

Kind regards, 

on behalf of

Dr. Mudassar Raza 

Academic Editor

PLOS ONE